# Stress-Tuned Optical Transitions in Layered 1T-MX_2_ (M=Hf, Zr, Sn; X=S, Se) Crystals

**DOI:** 10.3390/nano12193433

**Published:** 2022-09-30

**Authors:** Miłosz Rybak, Tomasz Woźniak, Magdalena Birowska, Filip Dybała, Alfredo Segura, Konrad J. Kapcia, Paweł Scharoch, Robert Kudrawiec

**Affiliations:** 1Department of Semiconductor Materials Engineering, Faculty of Fundamental Problems of Technology, Wrocław University of Science and Technology, Wybrzeże Wyspiańskiego 27, 50-370 Wrocław, Poland; 2Institute of Theoretical Physics, Faculty of Physics, University of Warsaw, Pasteura St. 5, 02-093 Warsaw, Poland; 3Departamento de Física Aplicada-ICMUV, Malta-Consolider Team, Universitat de València, 46100 Burjassot, Spain; 4Institute of Spintronics and Quantum Information, Faculty of Physics, Adam Mickiewicz University in Poznań, ul. Uniwersytetu Poznańskiego 2, 61-614 Poznań, Poland; 5Center for Free-Electron Laser Science CFEL, Deutsches Elektronen-Synchrotron DESY, Notkestr. 85, 22607 Hamburg, Germany

**Keywords:** MX_2_, DFT, bulk, band structure, pressure coefficients, transition metal dichalcogenides

## Abstract

Optical measurements under externally applied stresses allow us to study the materials’ electronic structure by comparing the pressure evolution of optical peaks obtained from experiments and theoretical calculations. We examine the stress-induced changes in electronic structure for the thermodynamically stable 1T polytype of selected MX${}_{2}$ compounds (M=Hf, Zr, Sn; X=S, Se), using the density functional theory. We demonstrate that considered 1T-MX${}_{2}$ materials are semiconducting with indirect character of the band gap, irrespective to the employed pressure as predicted using modified Becke–Johnson potential. We determine energies of direct interband transitions between bands extrema and in band-nesting regions close to Fermi level. Generally, the studied transitions are optically active, exhibiting in-plane polarization of light. Finally, we quantify their energy trends under external hydrostatic, uniaxial, and biaxial stresses by determining the linear pressure coefficients. Generally, negative pressure coefficients are obtained implying the narrowing of the band gap. The semiconducting-to-metal transition are predicted under hydrostatic pressure. We discuss these trends in terms of orbital composition of involved electronic bands. In addition, we demonstrate that the measured pressure coefficients of HfS${}_{2}$ and HfSe${}_{2}$ absorption edges are in perfect agreement with our predictions. Comprehensive and easy-to-interpret tables containing the optical features are provided to form the basis for assignation of optical peaks in future measurements.

## 1. Introduction

Among the large family of van der Waals (vdW) crystals, transition metal dichalcogenides (TMDs) have attracted a great deal of interest owing to their unique combination of direct band gap, significant spin–orbit coupling and exceptional electronic and mechanical properties, making them attractive for both fundamental studies and applications [1,2]. In particular, their semiconducting nature opens a door to potential optoelectronic, photonic and sensing devices such as light emitting diodes, microlasers, solar cells, transistors or light detectors [3,4,5,6].

Optoelectronic properties of vdW materials can be tuned by multiple external factors. One of them is an effective strain engineering. Recent theoretical and experimental reports have demonstrated flexible control over their electronic states via applying external strains [7,8,9]. For instance, applying an uniaxial tensile strain to monolayer of MoS${}_{2}$ may result in direct-to-indirect band gap transition [10], whereas applying a biaxial strain gives rise to a semiconductor-to-metal phase transition [11]. Meanwhile, the prominent mechanical strength of TMDs [12], compared with conventional 3D semiconductors, allows to use large strains for band structure engineering. For instance, combined studies by means of density functional theory (DFT) calculations and atomic force microscopy measurements have reported that the fracture stress of a freely suspended MoS${}_{2}$ [12,13] approaches the theoretical limit of this quantity for defect-free elastic crystal (one-ninth its Young’s modulus) [14]. In addition, numerous nondestructive optical techniques, including Raman, absorption, photoreflectance, and photoluminescence experiments, can be readily employed to quantitatively determine strain-tuned optical properties. In addition, high-pressure measurements are highly desirable for detailed band structure information as well as give useful benchmark to test DFT calculations. Such techniques also provide a direct way to probe interlayer interaction in the layered structures. In particular, recent experimental reports have demonstrated that the energies of various optical transitions in TMDs exhibit significant pressure dependence [15,16,17,18], which allows for the identification of the optical peaks, making them attractive for applications in pressure-sensing devices [19,20,21]. Generally, the unique mechanical flexibility and strength of TMDs make them an ideal platform for band gap engineering by strain, thus, enabling enhancement of their optical properties.

The chemical formula of hexagonal TMDs is MX${}_{2}$, where M stands for a transition metal element, and X is a chalcogene element (S, Se or Te). TMDs exhibit several structural polytypes of which two most common are trigonal prismatic (2H) and octahedral (1T) ones (see Figure 1). The difference between 2H and 1T polytypes can be viewed in different arrangement of atomic planes sequence within the monolayer. Namely, 2H polytype corresponds to an ABA arrangement, whereas 1T polytype is characterised by ABC sequence order [22]. Although 2H polytype of TMDs, based on Mo and W, have been extensively studied, the octahedral 1T MX${}_{2}$ compounds containing the M=Hf, Zr and Sn, X=S, Se elements have been less examined. The latter ones are indirect-gap semiconductors with band gaps ranging from visible to near-infrared wavelengths [23]. The earlier studies on 1T-MX${}_{2}$ compounds have predicted very high electron mobility and sheet current density in HfS${}_{2}$, superior to MoS${}_{2}$ [24,25], which makes ultrathin HfS${}_{2}$ phototransistors appealing for optoelectronics [26]. Thin SnSe${}_{2}$ flakes were shown to exhibit high photoresponsivity [27]. ZrS${}_{2}$ nanosheets were found suitable as anodes for sodium ion batteries [28]. These findings motivate further studies of electronic properties of 1T-MX${}_{2}$ crystals in 1L and bulk form. Despite some works reporting pressure evolution of Raman spectra [29,30,31], as well as X-ray diffraction and transport measurements [32], optical measurements under pressure are largely missing for 1T-MX${}_{2}$ compounds.

In this work, we systematically investigate the impact of external stress on the basic features of the band structure of MX${}_{2}$ (M=Hf, Zr, Sn; X=S, Se) in the 1T bulk polytype by DFT calculations. For each compound, we identify the dominant direct electronic transitions in BZ. As the structural anisotropy of in-plane and out-of plane directions in layered systems may result in different response to the strain, we study the evolution of the band structure upon applying stress types that are most frequently realized in experiments, i.e., compressive isotropic (hydrostatic), biaxial, and uniaxial stress. We quantify the energy trend for each transition between ambient and band gap closing pressure by determining the linear pressure coefficients. In addition, we examine the effect of light polarization for optically active direct transitions using dipole selection rules. Our predicted pressure coefficients and polarization of transitions can serve for identification of the features in measured optical spectra. Meanwhile, we explain the observed chemical trends by the orbital composition of electronic bands involved in the transitions. Finally, we compare our calculated results to the pressure trends of absorption edges positions measured in HfS${}_{2}$ and HfSe${}_{2}$ crystals, finding an excellent agreement. It corroborates that our adopted computational strategy is accurate at the quantitative level.

## 2. Methods and Materials

The DFT calculations have been performed in Vienna Ab Initio Simulation Package [33]. The electron-ion interaction was modeled using projector-augmented-wave technique [34]. In the case of tin (Sn) atom, the 4d${}^{10}$ states were included in valence shell, for hafnium and zircon, additional s states were taken (4s${}^{2}$ for Zr, 5s${}^{2}$ for Hf). The Perdew–Burke–Ernzerhof (PBE) [35] exchange-correlation (XC) functional was employed. A plane-wave basis cutoff of 500 eV and a 12×12×8 Monkhorst-Pack [36] k-point grid for BZ integrations were set. These values assured the convergence of the lattice constants and the electronic gaps were within precision of 0.001 Å and 0.001 eV, respectively. A Gaussian smearing of 0.02 eV was used for integration in reciprocal space. It is well known that standard exchange correlation functionals are insufficient to describe a non-local nature of dispersive forces, crucial to obtain a proper interlayer distance for layered structures [37,38]. Thus, the semi-empirical Grimme’s correction with Becke–Johnson damping (D3-BJ) [39] was employed to properly describe the weak vdW forces. The spin–orbit (SO) interaction was taken into account.

It is well established that the standard approximations to the XC functional lead to a severe underestimation of the electronic band gap and the lack of inclusion excitonic effects. In this regard, DFT is inaccurate for identification of optical transitions based on their absolute energy values. This issue can be partly improved by using more advanced techniques such as hybrid functionals or GW method [40,41], but their computational costs often make the calculations unfeasible for systems containing more than few atoms. The modified Becke–Johnson (mBJ) potential is an alternative approach to improve the band gaps with relatively low computational cost [42,43,44]. Recent report shave shown that mBJ provides reasonable results for identifying the optical transitions in ReS${}_{2}$ and ReSe${}_{2}$ bulk crystals [16]. It also yields pressure coefficients of optical transitions in excellent agreement with experimental values [15,45]. Therefore, we employ mBJ potential for band structure calculations, on top of the optimized geometry obtained within the PBE+D3-BJ+SO approach. The direct interband momentum matrix elements were computed from the wave function derivatives using density functional perturbation theory [46].

## 3. Results

### 3.1. Theoretical Analysis

We start our research by considering the geometry and electronic structure for the unstrained systems. The optimized lattice parameters, provided in Table 1, are in perfect agreement with experimental values. Similarly to 2H-TMDs, lattice constants are mostly governed by chalcogene atoms [15]. As it is expected for heavier atoms, the selenium (Se) compounds possess larger lattice parameters than sulfur (S) ones. The electronic band structures calculated under ambient conditions are presented in Figure 2. The band edges are located at the same high symmetry k-points for all studied systems. Namely, the valence band maximum (VBM) and conduction band minimum (CBM) are located at Γ and *L* k-points, respectively. Note that the VBM of SnSe${}_{2}$ at ambient pressure is not located exactly at Γ point, but between the Γ and K points (on the Γ-M path the local maximum is 2 meV lower). Under biaxial stress the VBM shifts to A point, but under hydrostatic and uniaxial stress the position and shape of VBM remain unchanged. This type of pressure behavior has already been observed in InSe crystals, where VBM exhibits toroidal shape [47,48]. The toroidal shape has consequences in transport and optical properties and would require further investigations, which are beyond the scope of our work. The calculated fundamental gaps exhibit indirect character with values systematically lower by 30–50% with respect to experimental values (see Table 1). The systems containing Se atoms exhibit reduced size of the energy gaps in comparison to S-containing systems.

The underestimation of the band gap is related to the geometrical structure—a better agreement is obtained with the use of experimental lattice constants, as shown in Ref. [49] and discussed in Appendix B for ZrSe${}_{2}$. In our study, we focus, however, not on the absolute value of the band gap, but rather on the pressure dependence of optical transitions, which requires a full optimization of geometry. Further, the discrepancies between theoretical and experimental bands gaps stem from the systematic underestimation of the band gap and the lack of including excitonic effects in our theoretical approach. On the other hand, the quasi-two-dimensional character of layered crystal leads to exciton binding energies on the order of tens or hundreds of meV [50,51,52,53,54], which redshifts the optical energies from their band-to-band values. Incidentally, it can improve the agreement with experiments, but this is fortuitous result and strongly material-dependent. In contrast to the absolute energy of transition, variation of its energy with respect to pressure, quantified by a linear pressure coefficient, demonstrates to be in good agreement with measured value [15,16,17]. Additionally, the dependence of the exciton binding energies upon the pressure can be neglected, whenever the exciton binding energy is much smaller than the transition energy [45,55]. Aforementioned suggest that the pressure coefficients obtained using mBJ might provide reasonable values and enable proper identification of the measured optical peaks on a quantitative level. Therefore, in order to compare the optical experimental results with our theoretical outcomes, the pressure coefficients are computed.

Although structural phase transitions under pressure were reported for our compounds [30,31,66,67,68,69], they are out of the scope of this work and we consider only the 1T phase under hydrostatic pressures up to metallization limit. In our mBJ-PBE+SO calculations they occur at pressures of: 266 kbar for SnS${}_{2}$, 188 kbar for HfS${}_{2}$, 128 kbar for ZrS${}_{2}$, 84 kbar for SnSe${}_{2}$, 65 kbar for HfSe${}_{2}$ and 26 kbar for ZrSe${}_{2}$. We also apply uniaxial and biaxial stresses, as depicted on Figure A2, which result from reducing the lattice parameters by up to 8% (see Appendix B). Figure 3 presents the band structures of HfS${}_{2}$ and SnSe${}_{2}$ under hydrostatic, uniaxial, and biaxial stress, as representatives of (Hf,Zr)X${}_{2}$ and SnX${}_{2}$ groups. Note, that the indirect character of the band gaps is preserved, irrespective of the pressure applied. The band edges positions are located at the same k-points as for unstrained samples, except for SnX${}_{2}$ systems under hydrostatic pressure, where CBM moves from L to M point, and biaxial strains, where VBM moves from Γ to A point. Note that, the application of compressive uniaxial strains result in reduction in the band gaps. Notably, the impact of hydrostatic or biaxial pressures is non-trivial and more complicated. In particular, for SnX${}_{2}$ compounds, the biaxial stresses initially increase the energy gap and move the VBM from Γ to A point (see Figure 3b and the Appendix B for a detailed discussion).

On the basis of the calculated electronic band structures, we identify direct electronic transitions with energies below 4.5 eV that might be optically active. For that, we calculate the energy differences between three uppermost valence bands (VB, VB-1, and VB-2) and three lowermost conduction bands (CB, CB+1, and CB+2) and plot them along the k-path, as presented in Figure 4c,d. The minima and plateaus in these plots denote the regions of BZ, where transitions occur between bands extrema at high-symmetry k-points or between parallel bands between high-symmetry k-points, called band-nesting regions. Both types of transitions contribute to van Hove’s singularities, that give rise to measurable optical signal. Figure 4c,d reveal multiple of such transitions in HfS${}_{2}$ and SnSe${}_{2}$. Additionally, in order to be optically active a transition must have a finite dipole strength, or intensity. Their calculated values are presented in Figure 4e,f, distinguishing the in-plane and out-of-plane components. In case the former (latter) one has a non-zero value, the emitted light propagates perpendicular (parallel) to the layers’ plane. Basing on these two conditions, we identify the optically active transitions in all studied compounds and depict the exemplary ones for HfS${}_{2}$ and SnSe${}_{2}$ with arrows in Figure 4a,b.

A significant difference between the systems containing transition metal (Hf, Zr) compared to the *p*-block metal (Sn) is visible in the orbital composition of their conduction bands. The lack of the *d*-type orbitals close to the Fermi level for Sn compounds substantially affects the curvature of the bands. Note that, for SnS${}_{2}$ and SnSe${}_{2}$ systems CBM is separated from other conduction bands by several hundred meV (see Figure 2a,b). Such separation reflects an intermediate character of the CBM, thus in turn, significantly decreases the number of optical transitions potentially visible in experiments. In particular, two transitions with highest intensities and the different polarization of light are located at Γ (from VB to CB, out-of plane polarization) and M (from VB to CB+1, in-plane polarization) k-points, respectively, (see Figure 4f,b). However, the latter one is less plausible due to much higher energy difference. Considering the Hf, Zr contained systems, their conduction bands are mainly composed of 4d states, and many bands appeared in CB, resulting in a higher number of plausible transitions (see Appendix D).

After selecting the direct transitions at high symmetry points and band-nesting regions with non-zero intensities, we discuss the pressure coefficients assigned to them and collected in Table 2. Note that, the transition energies exhibit nonlinear trends in the whole pressure range, as exemplified on Figure 5a,c for VB-CB transition in HfS${}_{2}$ and SnSe${}_{2}$. Thus, for each compound we divide the whole pressure range into three ranges, denoted I, II, and III, where the energy trends are approximately linear (for the complete list see Table A1). Figure 5b,d present the dispersion of the pressure coefficient for VB-CB transition in HfS${}_{2}$ and SnSe${}_{2}$. In HfS${}_{2}$, the pressure coefficients in II and III are similar and significantly different from I at Γ, K, L, and H points. In SnSe${}_{2}$ the differences are even larger. For example, the VB-CB transition at Γ exhibits a negative pressure dependence for low pressures (0–25 kbar). In range II (25–60 kbar) the energy is nearly constant and increases at higher pressures (>60 kbar).

The five lowermost optically active transitions for each compound, are collected in Table 2. The full list of all possible transitions up to 4.5 eV upon various stresses, with corresponding pressure coefficient are presented in Table A2, Table A3 and Table A4. Most of the selected transitions are located at high symmetry k-points, with a few band-nesting transitions present on Γ-A and A-L paths. Most of transitions exhibit in-plane polarization of the light. The Se-based compounds display higher pressure coefficients than S-based ones. In particular, for the lowermost optical transition the uniaxial coefficients have in the former ones have at least two times larger absolute values than in the latter ones.

The fundamental difference between the systems containing transition metal atoms and Sn atoms is observed in the sign of pressure coefficients. For the first ones, most of the predicted pressure coefficients are negative, whereas for the systems with Sn atoms, the positive values are obtained in the case of hydrostatic and biaxial stresses (see Figure 2 and Figure 6). In addition for SnX${}_{2}$ the largest pressure coefficients are obtained for uniaxial stresses (one order of magnitude larger than the rest ones), while for the rest structures it is not conclusive.

### 3.2. Comparison with Experiment

To benchmark our results, we compare them to the pressure trends of absorption edge positions measured in HfS${}_{2}$ and HfSe${}_{2}$. For that purpose the commercial HfS${}_{2}$ and HfSe${}_{2}$ crystals were used, grown using the state-of-art flux zone technique at 2D Semiconductors company. In order to measure the pressure dependence of absorption spectra, crystals were exfoliated to bulk-like flakes and enclosed in a diamond anvil cell (DAC). A helium-filled membrane was used to control the thrust on the diamonds and the pressure inside the chamber. The value of pressure was determined by measuring the red shift of R1 photoluminescence line from ruby spheres using the high resolution Ocean Optics HR2000+ fiber spectrometer with 1800 g/mm grating and a silicon CCD detector. Light from standard halogen lamp was focused onto the sample with reflective objectives. The absorption spectra from the sample were measured by a second spectrometer of the same type but equipped with 300 g/mm grating.

Figure 6a,b show the pressure dependence of absorption spectra measured at room temperature for HfS${}_{2}$ and HfSe${}_{2}$, respectively. In order to extract the absorption edge, the square of absorption was plotted and extrapolated to zero. The obtained pressure dependence of absorption edge is plotted by solid circles in Figure 6c,d for HfS${}_{2}$ and HfSe${}_{2}$, respectively, together with the linear fit. The respective pressure coefficients are equal to −5.13 and −7.89 meV/kbar. According to theoretical calculations the observed absorption edge can be attributed to the direct optical transition at the Γ point of Brillouin zone. The pressure coefficient for this transition in HfS${}_{2}$ and HfSe${}_{2}$ determined from DFT are equal, respectively, to −5.49 and −7.34 meV/kbar. Note, that these theoretical values are different than those collected in Table 2, due to different pressure ranges for linear fitting. The experimental and theoretical values are in excellent agreement, confirming a quantitative accuracy of our calculations. Basing on that we predict, with high level of credibility, that it applies also to other compounds, not investigated experimentally yet.

## 4. Summary

This work reports an extensive DFT investigation of band structure evolution and electronic features upon applying various pressures for the bulk MX${}_{2}$ compounds (M=Hf, Zr, Sn; X=S, Se) in 1T polytype. We study the trends of the fundamental indirect band gap and direct transitions energies upon application of hydrostatic, uniaxial and biaxial stresses, up to semiconductor-to-metal transition. We provide the values of pressure coefficients in the ranges of linear behavior of transitions energies. Additionally, dipole strengths and polarizations of direct transitions are computed. The observed chemical trends are discussed in terms of orbital composition of involved electronic bands. In general, the negative pressure coefficients have been determined, except the (M=Sn; X=Se, S) structures under hydrostatic and biaxial stresses. The largest pressure coefficients are predicted under uniaxial stresses for Sn containing structures. We compare the calculated pressure coefficients to the experimental values for the absorption edges of HfS${}_{2}$ and HfSe${}_{2}$, obtaining and excellent agreement. It corroborates, that our computational strategy (PBE+D3-BJ+SO for geometry optimization, mBJ+SO for band structure calculations), yields a quantitative precision for identification of optical peaks in 1T-MX${}_{2}$, based on their pressure evolution. Our work provides easy-to-interpret tables with electronic band structure features under applying stress. Such results provide an indispensable and complete aid for materials characterization, as they enable assignment of measured optical peaks to specific transitions in the electronic band structure.

## Figures and Tables

**Figure 1 nanomaterials-12-03433-f001:**
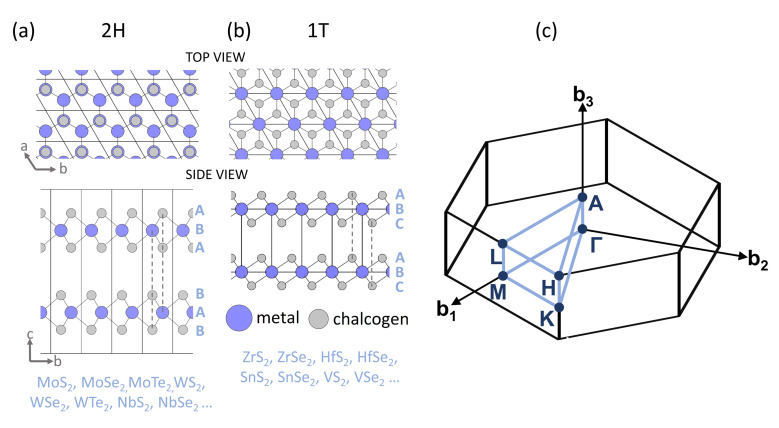
Top and side views of the (**a**) trigonal prismatic (2H) and (**b**) octahedral (1T) polytypes of MX${}_{2}$. (**c**) The first Brillouin zone (BZ) with high-symmetry k-points and lines denoted in blue.

**Figure 2 nanomaterials-12-03433-f002:**
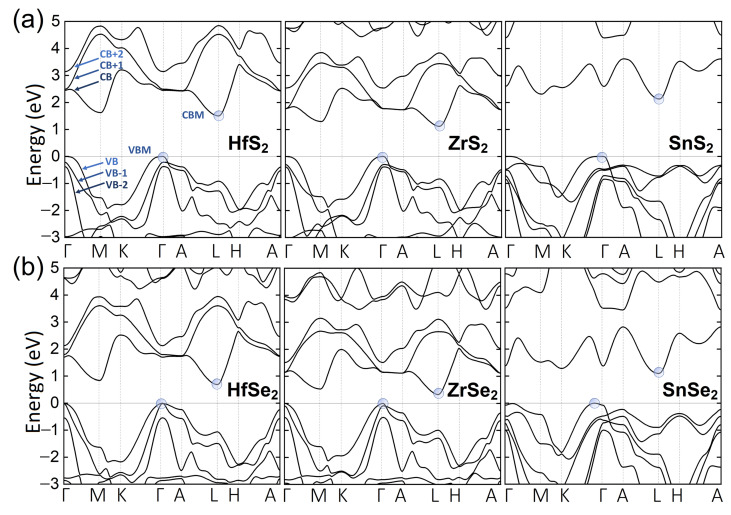
The electronic structure of bulk MS${}_{2}$ (M=Hf, Zr, Se) for (**a**) and MSe${}_{2}$ for (**b**) high symmetry lines in BZ obtained using mBJ potential on the top of PBE+D3-BJ+SO geometry optimization. The VBM and CBM are denoted in blue circles.

**Figure 3 nanomaterials-12-03433-f003:**
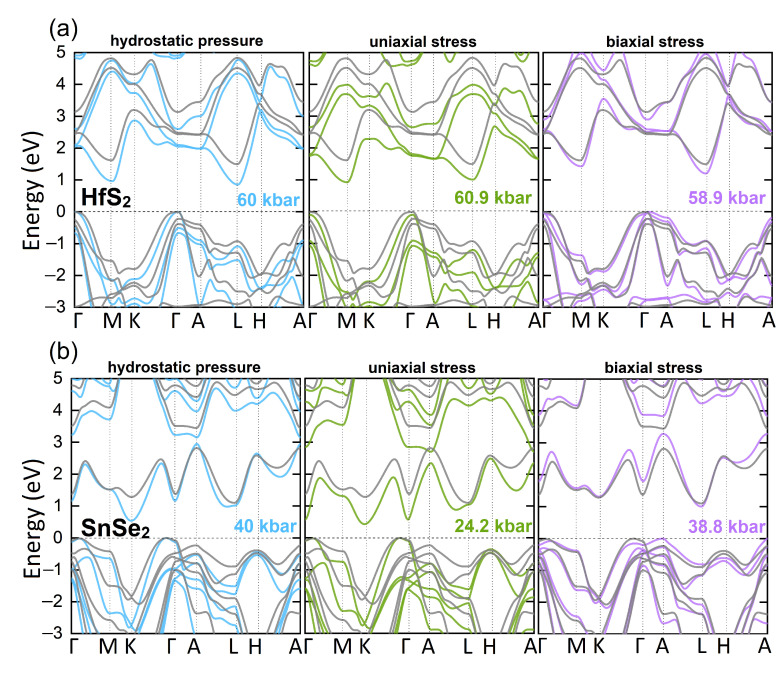
The electronic band structure for unstressed (grey lines) and stressed samples under hydrostatic pressure (blue lines), uniaxial stress (green lines), biaxial stress (violet lines), respectively, for (**a**) HfS${}_{2}$ and (**b**) SnSe${}_{2}$. The uniaxial and biaxial stresses correspond to −8% and −2% strains, respectively. Zero in energy is rigidly set to the VBM.

**Figure 4 nanomaterials-12-03433-f004:**
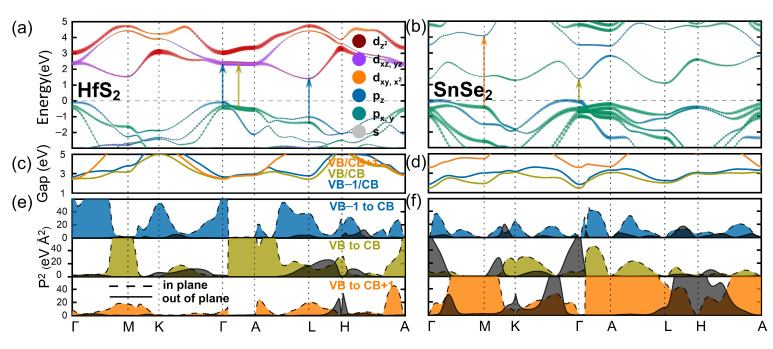
Electronic band structures with orbital projections for (**a**) HfS${}_{2}$, (**b**) SnSe${}_{2}$. The arrows in (**a**,**b**) mark the optically active transitions with highest intensities. The color of the arrow reflects the position from which the transition occurs. For instance, the blue arrow denotes the transition from VB-1 to CB, whereas the light green—from VB to CB. (**c**,**d**) graphs show energy of direct transition between selected bands (the other pairs of bands are omitted for the sake of clarity), and (**e**,**f**) demonstrate the corresponding dipole strengths and polarizations of transitions.

**Figure 5 nanomaterials-12-03433-f005:**
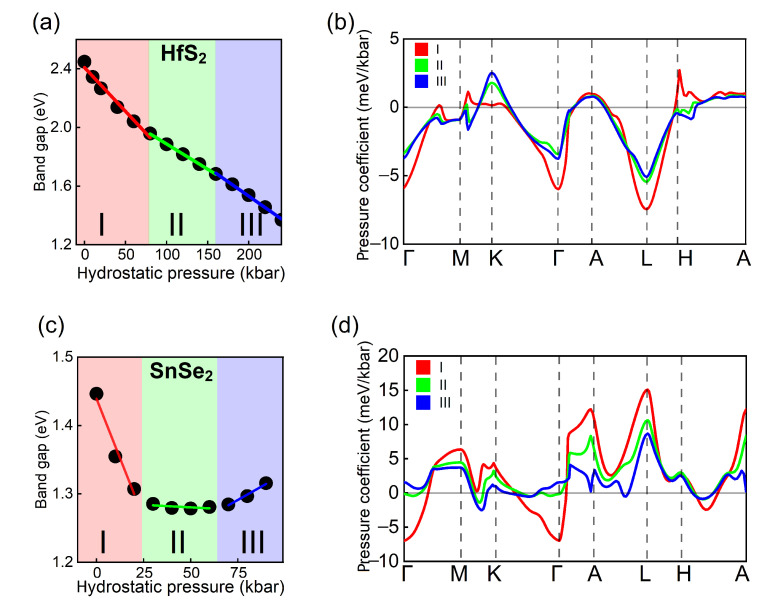
The energy transition under the hydrostatic pressure at Γ point for (**a**) HfS${}_{2}$ (**c**) SnSe${}_{2}$ bulk structures. Three linear ranges have been selected (I, II, III), different for both systems. Corresponding pressure coefficients for HfS${}_{2}$ and SnSe${}_{2}$, are presented in (**b**,**d**), respectively, along the entire k-path.

**Figure 6 nanomaterials-12-03433-f006:**
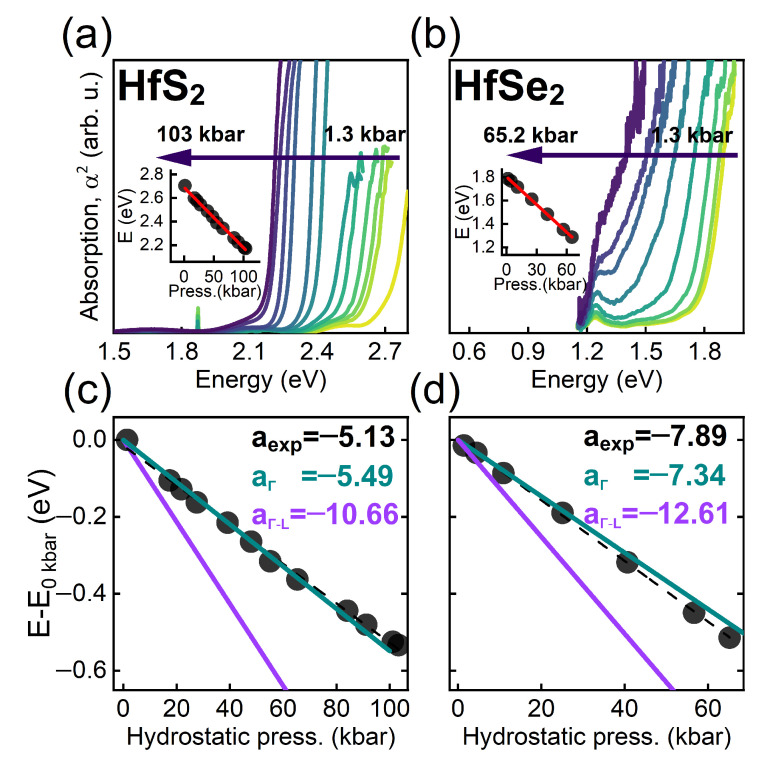
Pressure dependence of absorption spectra measured at room temperature for (**a**) HfS${}_{2}$ and (**b**) HfSe${}_{2}$. Inset shows the absorption edge determined from the absorption spectra (solid circles) together with the linear fit (red lines). Pressure dependencies of the absorption edge energy (black points) and calculated VB-CB transition at Γ (green) and fundamental indirect transition Γ-L (violet) as a linear fit of points on the same range and derived pressure coefficients (meV/kbar) for (**c**) HfS${}_{2}$ and (**d**) HfSe${}_{2}$.

**Table 1 nanomaterials-12-03433-t001:** Calculated and measured lattice constants and fundamental band gaps of all the compounds.

System	$a^{\mathrm{DFT}}$ (Å)	$c^{\mathrm{DFT}}$ (Å)	$E_{g}^{\mathrm{DFT}}$ (eV)	$a^{\exp}$ (Å)	$c^{\exp}$ (Å)	$E_{g}^{\exp}$ (eV)
HfS${}_{2}$	3.59	5.75	1.50	3.63 [56]	5.86 [56]	1.96 [57], 1.80 [58], 1.87 [59]
HfSe${}_{2}$	3.70	6.08	0.71	3.67 [56]	6.00 [56]	1.13 [57], 1.15 [58]
ZrS${}_{2}$	3.63	5.72	1.12	3.66 [57]	5.82 [57]	1.68 [57], 1.70 [60], 1.78 [61]
ZrSe${}_{2}$	3.74	6.04	0.33	3.77 [57]	6.14 [57]	1.20 [60], 1.10 [62], 1.18 [63]
SnS${}_{2}$	3.67	5.80	2.14	3.65 [64]	5.90 [64]	2.88 [65]
SnSe${}_{2}$	3.84	6.00	1.10	3.82 [64]	6.14 [64]	1.63 [65]

**Table 2 nanomaterials-12-03433-t002:** Direct optical transitions for particular high symmetry k-points for all employed compounds. The n.XY represents the nesting bands transition between the X and Y k-points, whereas the *v* (valence band) and *c* (conduction band) indicate positions for which the transition occurs. The *E*, $P_{\|}^{2}$ and $P_{⊥}^{2}$ denote the energy of transition, intensity of in-plane and out-of plane polarization of light (given in $eV^{2}$Å${}^{2}$), respectively. The last nine columns indicate the pressure coefficients (given in meV/kbar), where the lower index denotes the pressure regions defined in Appendix C, and the upper index indicates the type of pressure: h—hydrostatic, u—uniaxial, and b—biaxial.

Point	v	c	E	$P_{\|}^{2}$	$P_{⊥}^{2}$	$a_{I}^{h}$	$a_{\mathrm{II}}^{h}$	$a_{\mathrm{III}}^{h}$	$a_{I}^{u}$	$a_{\mathrm{II}}^{u}$	$a_{\mathrm{III}}^{u}$	$a_{I}^{b}$	$a_{\mathrm{II}}^{b}$	$a_{\mathrm{III}}^{b}$
						${\mathrm{HfS}}_{2}$								
L	VB	CB	2.076	21	11	−7.61	−5.45	−5.05	−5.16	−2.91	−1.72	−5.49	−6.99	−7.68
Γ	VB	CB	2.448	0	3	−6.64	−3.56	−3.78	−12.83	−9.51	−8.00	1.08	-6.29	−9.26
n.ΓA	VB	CB	2.687	67	0	−4.19	−1.71	−0.96	−9.16	−7.96	−7.03	−1.71	−2.07	−2.98
Γ	VB-1	CB	2.690	66	0	0.87	0.66	0.66	−0.21	0.14	0.25	−1.63	−6.74	−9.75
L	VB-1	CB	2.781	30	2	−2.63	−1.26	−0.94	1.06	2.63	3.50	−6.89	−4.89	−4.98
						${\mathrm{HfSe}}_{2}$								
Γ	VB	CB	1.699	52	0	−9.04	−6.57	−5.79	−10.77	−8.70	−7.94	−8.15	−13.52	−14.72
Γ	VB-1	CB	1.716	5	5	−4.02	−4.06	−4.79	0.84	1.03	1.13	−4.34	−12.03	−13.93
L	VB	CB	1.758	14	14	−10.18	−8.11	−6.97	−3.32	−2.02	−1.26	−8.63	−9.98	−9.97
Γ	VB-1	CB+1	1.811	22	0	−2.92	−2.33	−1.96	0.34	0.71	0.94	0.40	−2.10	−2.58
A	VB	CB	1.962	68	0	1.32	1.02	0.79	4.02	3.63	3.57	−3.06	−3.65	−4.49
						${\mathrm{ZrS}}_{2}$								
Γ	VB	CB+1	1.777	4	0	−5.41	−2.53	−1.69	−11.51	−8.27	−7.22	−3.73	−3.09	−2.86
L	VB	CB	1.949	10	20	−7.64	−5.70	−5.30	−5.17	−2.70	−1.52	−1.68	−1.01	−0.60
Γ	VB-1	CB+2	2.073	11	0	−4.41	−4.44	−4.10	3.65	4.20	4.69	1.18	1.57	1.86
Γ	VB-2	CB+1	2.164	41	0	−1.37	−0.96	−0.78	−0.16	0.23	0.40	−0.05	0.09	0.16
A	VB	CB	2.209	61	0	1.36	1.11	1.13	2.96	2.67	2.63	0.96	1.00	1.04
						${\mathrm{ZrSe}}_{2}$								
Γ	VB	CB+1	1.151	14	0	−6.76	−3.50	−2.46	−8.29	−5.75	−4.92	−2.92	−3.48	0.60
n.ΓA	VB	CB	1.252	54	0	−5.31	−2.36	−1.46	−10.08	−7.87	−7.06	−3.62	−10.30	−12.88
Γ	VB-1	CB	1.252	53	0	−2.87	−2.88	−5.39	1.34	1.38	1.34	−1.65	−12.33	−9.91
L	VB	CB	1.320	23	6	−10.03	−7.75	−7.10	−3.25	−1.86	−0.98	−9.22	−10.43	−10.32
A	VB	CB	1.460	60	0	1.64	1.33	0.89	4.46	3.91	3.87	−2.94	−3.10	−3.34
						${\mathrm{SnS}}_{2}$								
Γ	VB	CB	2.595	0	61	−4.73	0.22	1.28	−28.59	−25.78	−24.78	13.08	1.03	−1.09
L	VB-2	CB	3.786	3	35	2.20	4.21	8.71	−4.56	−1.89	−0.30	4.32	1.38	−2.14
H	VB	CB	3.871	4	0	3.01	3.00	2.85	−2.63	−3.11	−3.71	3.84	2.21	1.42
H	VB-1	CB	3.889	7	0	2.89	3.01	2.85	−2.48	−2.99	−3.60	3.78	2.16	1.38
K	VB	CB	3.953	20	0	3.87	2.52	1.65	1.38	0.96	0.55	2.64	1.52	0.96
						${\mathrm{SnSe}}_{2}$								
Γ	VB	CB	1.446	8	67	−5.31	−0.15	1.17	−34.07	−31.48	−32.20	11.54	−0.14	−2.98
Γ	VB-2	CB	1.959	112	0	3.18	8.11	8.06	−6.45	−5.74	2.35	4.45	0.08	−2.75
H	VB	CB	2.969	3	3	2.74	3.00	2.56	−4.41	−5.21	−6.68	4.16	2.25	1.21
L	VB-2	CB	2.986	14	14	4.16	4.95	4.96	1.36	5.21	8.35	1.66	−2.95	−4.30
K	VB	CB	3.072	22	0	4.17	3.24	1.43	0.47	−0.52	−1.76	2.73	1.42	0.69

## Data Availability

The data presented in this study are available on request from the corresponding authors.

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
