# Peer review of "Stress-Tuned Optical Transitions in Layered 1T-MX2 (M=Hf, Zr, Sn; X=S, Se) Crystals"

_nanomaterials, 2022, doi:10.3390/nano12193433_

Round 1

Reviewer 1 Report

This manuscript reports the density functional theory study of the stress-induced changes in electronic structure for the thermodynamically stable 1T polytype of selected MX2 compounds (M = Hf, Zr, Sn; X = S, Se). The manuscript is well organized and worthy of publication. But there are still some issues needed to be addressed before publication. The detailed comments are as following,

1.     In previous literatures, the maximum pressures have been up to tens of GPa, and many interesting phenomena, such as structural and electronic phase transitions, have been observed. Why the maximum pressure was just up to 200 kbar in this work? In this manuscript, only the shift of band gap was reported under low stress. Have the authors examined the changes the electronic structure of selected MX2 under high pressure?  

2.     There are some small typos in the manuscript. For example, Page 6, Line 172, “Figs. 4(e, ff)”. Page 7, Line 194, “denoted I, III and III,”

Author Response

Response to Reviewer 1 Comments

Below, we provide our detailed responses to the Referee’s comments. For clarity, the Referee’s comments are given in blue colored text, our replies in black, and changes in the revised manuscript are indicated by red color. All the references to bibliography positions, figures and lines numbers correspond to the revised version of the manuscript.

This manuscript reports the density functional theory study of the stress-induced changes in electronic structure for the thermodynamically stable 1T polytype of selected MX2 compounds (M = Hf, Zr, Sn; X = S, Se). The manuscript is well organized and worthy of publication. But there are still some issues needed to be addressed before publication. The detailed comments are as following,

We are very grateful for the Referee’s positive evaluation of our manuscript. Below, we carefully address the Referee’s comments.

Point 1: In previous literatures, the maximum pressures have been up to tens of GPa, and many interesting phenomena, such as structural and electronic phase transitions, have been observed. Why the maximum pressure was just up to 200 kbar in this work? In this manuscript, only the shift of band gap was reported under low stress. Have the authors examined the changes the electronic structure of selected MX2under high pressure?  

Response 1: As emphasized in lines 151-153 of the manuscript, the maximal hydrostatic pressure was set to the value at which the band gap closes in our PBE + D3 mBJ-LDA calculations. We are aware of the structural and electronic phase transitions reported in references [68-73] for these compounds at higher pressures. However, in our work we focused only on 1T phase.

Point 2: There are some small typos in the manuscript. For example, Page 6, Line 172, “Figs. 4(e, ff)”. Page 7, Line 194, “denoted I, III and III,”

Response 2: We thank the Referee for pointing out the typos. We  have carefully reviewed the manuscript and corrected all of them.

Reviewer 2 Report

In this work, the author used DFT methods systematically investigate the impact of external stress on the band structure of MX2 (M= Hf, Zr, Sn; X= S, Se). It is interesting and the results may provide aid for materials characterization. I think this paper should be published after minor revision in the following points:

1.        In the title of the article, the author's research objective is 1T-mX2 (M= Hf, Zr, Sn; X= S, Se) six compounds. The electronic band structures of all the six compounds are showed in Fig 2. Why were only two compounds (HfS2 and SnSe2) compared for stress and orbital projections. Most of the discussion in this paper focuses on just these two compounds. While when comes to 3.2 comparison with experiment, the subjects were changed again to HfS2 and HfSe2. I don't think it's helpful to make the point and fit the title.

2.        On line 180-181 “Note that, for SnS2 and SnSe2 systems CBM is separated from other conduction bands by several hundred meV (see 4(a, b)).” If the author is referring to Fig 4(a, b), no SnS2 was mentioned in this Figure.

3.        In Table 1. The EDFTg of ZrSe2 are very different from Eexpg compare with other compounds. What makes such a big difference?

4.        There are many details in the article that need to be corrected. Such as “Figs. 4 (e,ff)” on line 174, and lack of “Fig.” on line 186 “(see 4(f, b))”.

Reviewer 3 Report

Comments to the Author

Manuscript ID: nanomaterials-1920325

Title: Stress-tuned optical transitions in layered 1T-MX2 (M= Hf, Zr, Sn; X= S, Se) crystals.

The work presented by Rybak and co-workers is interesting and impacts to appeal wide readership of inorganic chemists. The manuscript is overall well-structured and well documented. Authors have demonstrated computation studies of band structure evolution and electronic features upon applying various pressures for the compounds MX2 where (M = Hf, Zr, Sn; X = S, Se) in 1T phase. The DFT investigation is done thoroughly and is satisfactory with experimental results. I recommend this manuscript for publication in Nanomaterials in its present format.
